# Suitability and user acceptance of the eResearch system "Prospective Monitoring and Management App (PIA)"—The example of an epidemiological study on infectious diseases

**Julia Ortmann**[1‡], **Jana-Kristin Heise**[1,2‡], **Irina Janzen**[1], **Felix Jenniches**[1], **Yvonne Kemmling**[1], **Cornelia Frömke**[3], **Stefanie Castell**[1,2]*

1 Department for Epidemiology, Helmholtz Centre for Infection Research, Brunswick, Lower Saxony, Germany, 2 German Centre of Infection Research (DZIF), Brunswick, Lower Saxony, Germany, 3 Hannover University of Applied Sciences and Arts, Hanover, Lower Saxony, Germany

‡ JO and JKH contributed equally to this work and are joint first authors.
* Stefanie.Castell@helmholtz-hzi.de

## Abstract

### Background

The eResearch system "Prospective Monitoring and Management App (PIA)" allows researchers to implement questionnaires on any topic and to manage biosamples. Currently, we use PIA in the longitudinal study ZIFCO (Integrated DZIF Infection Cohort within the German National Cohort) in Hannover (Germany) to investigate e.g. associations of risk factors and infectious diseases. Our aim was to assess user acceptance and compliance to determine suitability of PIA for epidemiological research on transient infectious diseases.

### Methods

ZIFCO participants used PIA to answer weekly questionnaires on health status and report spontaneous onset of symptoms. In case of symptoms of a respiratory infection, the app requested participants to self-sample a nasal swab for viral analysis. To assess user acceptance, we implemented the System Usability Scale (SUS) and fitted a linear regression model on the resulting score. For investigation of compliance with submitting the weekly health questionnaires, we used a logistic regression model with binomial response.

### Results

We analyzed data of 313 participants (median age 52.5 years, 52.4% women). An average SUS of 72.0 reveals good acceptance of PIA. Participants with a higher technology readiness score at the beginning of study participation also reported higher user acceptance. Overall compliance with submitting the weekly health questionnaires showed a median of 55.7%. Being female, of younger age and being enrolled for a longer time decreased the odds to respond. However, women over 60 had a higher chance to respond than women

research data of NAKO Gesundheitsstudie. Data are available via the NAKO Gesundheitsstudie (contact via https://nako.de/) upon request for researchers who incl their proposal meet the criteria for access to these data.

**Funding:** This project was conducted with data from the German National Cohort (NAKO) (www. nako.de). The NAKO is funded by the Federal Ministry of Education and Research (BMBF) (project funding reference numbers: 01ER1301A/B/C and 01ER1511D), federal states and the Helmholtz Association with additional financial support by the participating universities and the institutes of the Leibniz Association. The Level 3 study ZIFCO is supported by the DZIF (German Centre for Infection Research) and the Helmholtz Association. The funders had no role in the study design, data collection and analysis, decision to publish, or preparation of the manuscript.

**Competing interests:** The authors have declared that no competing interests exist.

under 60, while men under 40 had the highest chance to respond. Compliance with nasal swab self-sampling was 77.2%.

## Discussion

Our findings show that PIA is suitable for the use in epidemiologic studies with regular short questionnaires. Still, we will focus on user engagement and gamification for the further development of PIA to help incentivize regular and long-term participation.

## Introduction

Acute respiratory infections (ARI) represent a large burden of disease in Germany each year [1, 2] but are difficult to study in population-based studies, in part because of their sudden onset as well as transient and nonspecific course. The current SARS-CoV-2 pandemic highlights this challenge and the importance on epidemiological research on such diseases. Digital tools as an alternative to paper-based reporting help to avoid recall bias through active symptom monitoring and improve response and compliance in epidemiological studies [3–6]. Together with the self-collection of biospecimens such as nasal swabs, they can thereby contribute to resolving methodological limitations [7–9]. The eResearch system "Prospective Monitoring and Management App (PIA)" consists of a mobile and a web-based application [10, 11]. The goal is to simplify and modernize the collection of epidemiological data on ARI and other infectious diseases by enabling implementation of surveys on different health topics, management of biospecimens, and rapid contacting of study participants. PIA is used within "Integrated DZIF Infection Cohort within German National Cohort (ZIFCO)" to investigate associations between risk factors and transient infectious diseases, e.g. respiratory, gastrointestinal, and urogenital infections. To ensure successful long-term use of PIA, as necessary for e.g. the investigation of seasonal diseases, good compliance of participants over time is required. For this, an application has to offer good usability [12–14]. As a generic survey tool, PIA can be used flexibly in different research contexts, as a stand-alone tool or integrated in other systems such as NatCoEdc which is the IT system of the German National Cohort (NAKO) [15]. In addition, PIA can be used as a symptom diary for contacts of COVID-19 cases and such cases themselves integrated into the software SORMAS (Surveillance Outbreak Response Management and Analysis System) that supports health departments in managing the COVID-19 pandemic [16]. Because of the use of PIA in various research settings and projects, findings on user acceptance of PIA and compliance within ZIFCO may have important implications for the future development of PIA. Therefore, our aim was to assess user acceptance of PIA and compliance over time to determine its suitability in research on transient infectious diseases.

## Methods

### Ethics statement

The internal data protection officer of the HZI approved a study specific data protection concept. In addition, the assessment of the German Federal Commissioner for Data Protection and Freedom of Information found no data protection concerns about the study ZIFCO in the version presented. The ethics committee of the Medical Association of Lower Saxony (Ärztekammer Niedersachsen) in Hannover had no objections to the implementation of the study (DRKS00021077).

## Study design and population

ZIFCO is an add-on study of the longitudinal population-based cohort study NAKO [15] and started recruiting on 18 October 2019. The NAKO includes a random sample of more than 200,000 men and women aged 20 to 69 at baseline and is carried out at 18 study centers throughout Germany [17]. ZIFCO includes long-term symptom monitoring of infectious diseases with PIA, e.g. respiratory, gastrointestinal and urogenital infections as well as potential risk factors. If written informed consent has been given, study nurses register participants in PIA and explain how to use the app, which is available for Android, iOS or as a web version [11]. Until 7 August 2020, the mobile app was only available through side loading. Participants also receive a nasal swab self-sampling kits to detect pathogens in case of a respiratory infection. After completing the study center visit, participants can use the application to report symptoms of infectious diseases for as long as they are willing to participate, ideally over years. As an incentive to increase compliance, participants can view results of the laboratory analysis of their nasal swabs in PIA. Withdrawal from the study as well as deletion of study data is possible at any time. A study hotline and support via e-mail are available in case of problems.

## App design, questionnaires, and data on user activity

The most important component of PIA in ZIFCO is the weekly health questionnaire that queries the health status and is presented each Monday. Participants can also report a spontaneous onset of disease at any time and enter their symptoms. If participants report symptoms of a new infection (respiratory, gastrointestinal, or urogenital), they are asked to provide further details. In case of acute respiratory infections (ARI) symptoms, they are additionally requested to self-sample a nasal biospecimen and send the sample for laboratory analysis of common respiratory viruses free of cost. Additional questionnaires investigate relevant factors for the course of e.g. ARI, for example vaccination, allergies, and contact with children.

At the beginning, participants are asked to provide data on demography (year of birth and gender). Since earlier studies showed a link between attitudes towards technology and user behavior of apps [14, 18], we also implemented a questionnaire that assesses technology readiness of participants two days after first log in. The questionnaire uses a twelve-item Likert scale on the three subtopics technology acceptance, technology competence and technology control that are combined to a technology readiness score [18]. To evaluate user acceptance of PIA, four months after registration users can answer a questionnaire based on the System Usability Scale (SUS) [19] containing a ten-item Likert scale.

Participants are asked to answer all questionnaires but may skip the ones, or items, they do not want to answer. To remind app users of participation, they receive push notifications or e-mails, depending on whether they last used the mobile or web version. Data on user activity including type of activity, timestamp, and which app system participants used (Android, iOS, web), are collected per default for the study ZIFCO but can be turned off by participants.

## Definitions and statistical analysis

The primary aim of our study was to investigate I) the acceptance of PIA and II) the compliance of users with submitting weekly health questionnaires regardless of their health status. For that purpose, we considered all data collected between 18 October 2019 and 31 March 2021 including demography, technology readiness and user acceptance as well as selected data on app use. Of secondary interest was compliance with submission of nasal swabs in case of respiratory symptoms. We only included participants in our analyses who provided information on age and gender.

Age was calculated by subtracting the birth year from the year of data analysis, and age groups were defined to investigate age group-specific pattern. We graphically compared the study population to the general population of Hannover, Germany, from which it was selected, in terms of gender and age group. We calculated the technology readiness score in accordance with the definition by Neyer et al. (2012) including all technology readiness questionnaires that users submitted without missing values. The resulting score ranges from 1 to 5; the higher the score, the higher technology readiness [18]. The SUS was calculated in accordance with Brooke [19] for all questionnaires without missing values. This score ranges from 0 to 100 and a mean score of greater or equal 71.4 is considered good user acceptance [20]. To investigate user behavior in general, we counted submissions of any questionnaire within PIA during our investigation period and examined the used app system (Android, iOS, web-app). However, to investigate compliance we only considered submission of the weekly health questionnaire as relevant, as this is the main questionnaire of ZIFCO. Compliance is therefore defined as the percentage of the number of answered weekly health questionnaires divided by the number of weeks since first registration in PIA, excluding weeks where PIA was not available due to technical problems (two weeks in study period). Since there is no standard definition of good compliance in the literature, we formed quartiles and considered compliance of at least 75% as good. Compliance with nasal swab self-sampling was defined as the percentage of all samples sent in divided by the number of samples that were requested to be send in. This request is automatically triggered in-app by certain answers in the symptom questionnaire (e.g. indication of fever, cough or runny nose).

Depending on the scale of the outcome, we fitted a linear regression model for user acceptance assessed with the SUS and a logistic regression model with binomial response for compliance. We included the technology readiness score as a predictor for user acceptance as well as for compliance with response. Other independent variables included age group, gender and app system used. Including age as a quadratic term in the models did not represent the results of exploratory analysis due to a non-linear nature between age and compliance; therefore, we used age groups as a predictor. For the logistic regression with binomial response, we additionally included the duration of enrollment (weeks since registration) as a predictor variable, but not the SUS because only participants that were enrolled for at least four months were able to answer the questionnaire assessing the SUS. Variable selection was done by conducting uni-variable models and including variables significant to a level of $\alpha = 20\%$ (two-sided) in the multivariable models. Additionally, we took association patterns already found in the exploratory analysis into account, as well as compared AIC, R2 and Log Likelihood Ratio. Variables in the multivariable models were considered significant with $p < 0.05$ (two-sided).

We excluded missing values for calculating proportions and stratifying descriptive statistics, as well as for the linear regression and binomial regression model.

We used R version 4.0.3 for data transformation and analysis.

## Results

### Participation

Between study start on 18 October 2019 and date of analysis 31 March 2021, 459 out of the 1,190 NAKO participants that completed the NAKO program at the study site enrolled in ZIFCO (38.6%). Due to organizational or technical circumstances, not all of the 1,190 NAKO participants were asked to participate ZIFCO.

Study nurses recruited individuals during their participation in the standard protocol of the NAKO. Nearly 70 percent (68.2%) of enrolled individuals for ZIFCO submitted the initial demography questionnaire (n = 313), which was mandatory for further participation (Fig 1), of which two did not answer the questionnaire completely (0.6%).

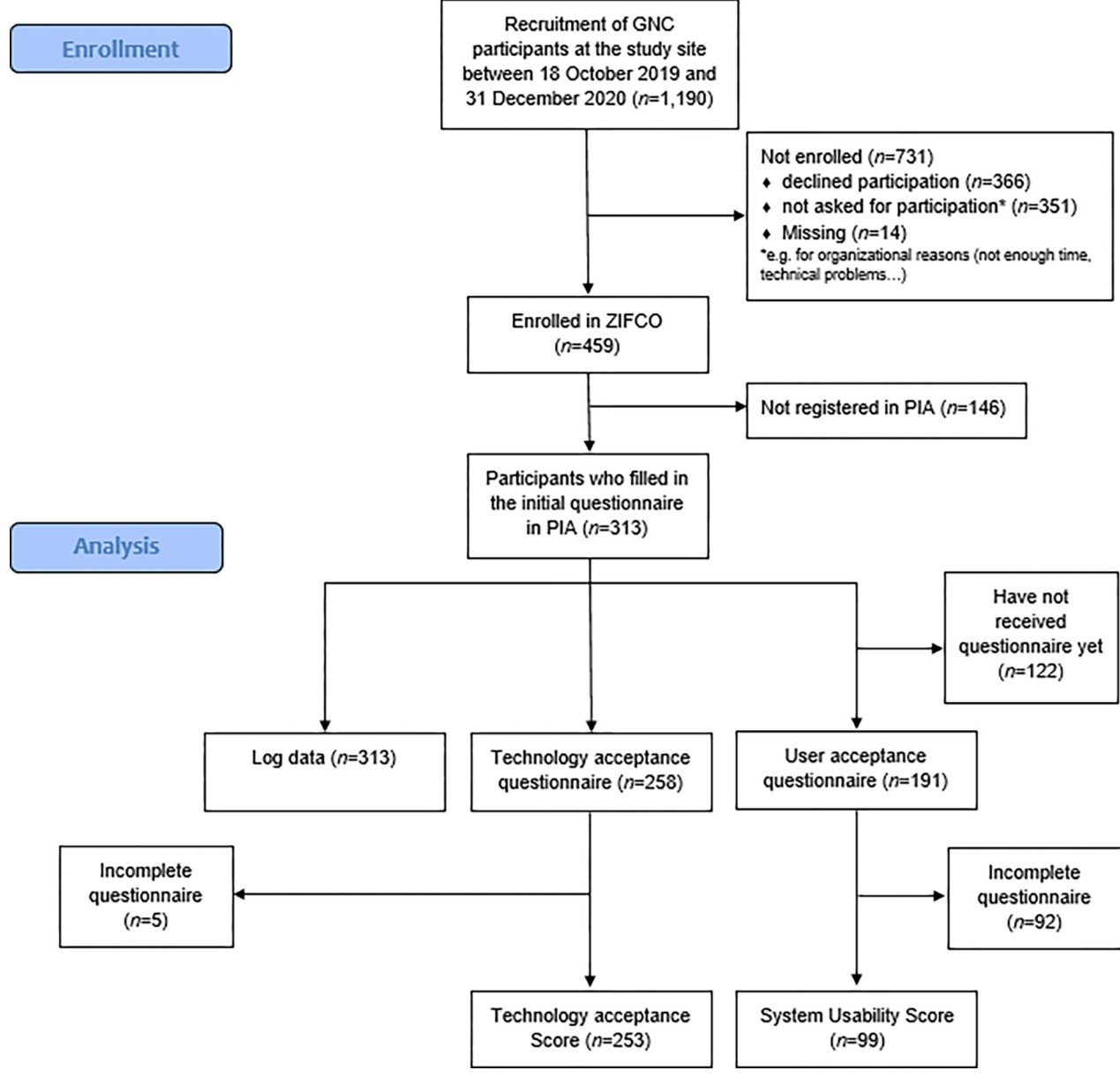

**Fig 1. Flow chart of participant enrollment and data analysis.**

Enrollment peaked in March 2021. During the lockdown and subsequent restrictions in Germany in response to the COVID-19 pandemic, the study center closed from 16 March to 26 June 2020 and from 16 December 2020 until the end of the year. During this time, there were no new participants (Fig 2).

Median age of participants was 52.5 years, 52.4% were female (163/311) (Table 1). Two participants did not provide data on gender or age. The study population of ZIFCO was older than the population of Hannover, Germany (Fig 3).

## User behavior

Eighty-two percent of users (258/313) used PIA only as a mobile app on an Android or iOS device to submit questionnaires respectively, while about 10% (30/313) used the web version

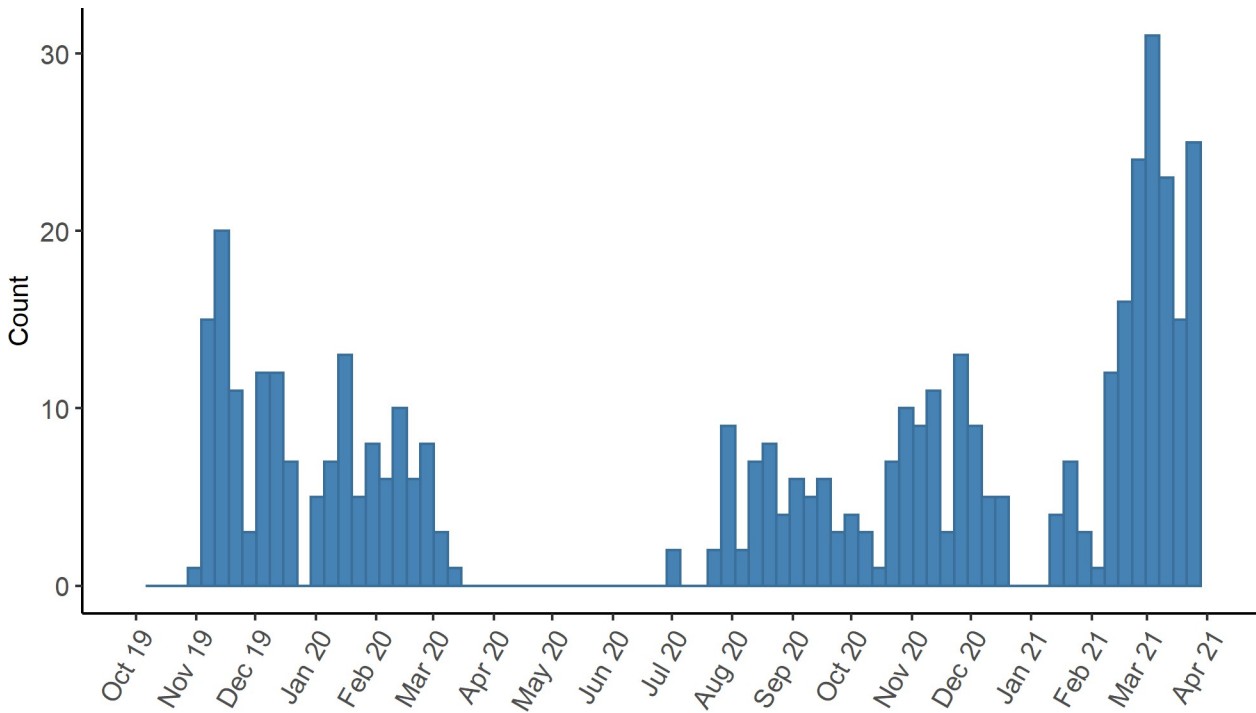

**Fig 2. Timeline of participant enrollment (18 October 2019–31 March 2021, *n* = 313).** Displayed are participants who submitted at least the initial questionnaire within PIA.

and 8% (25/313) changed the system they used, of which most changed between Android and web. Men over 60 years had the highest proportion of web version users (Table 2).

## Technology readiness

Eighty-one percent of participants (253/313) answered the technology readiness questionnaire completely resulting in a median technology readiness score of 2.9 out of possible 5 (IQR: 2.7–3.1) (S1 and S2 Tables).

## User acceptance

There were 191 participants that used the app for at least four months and were thus eligible to answer the questionnaire on user acceptance, of which about half completed the questionnaire (51.8%; 99/191). Those who did not fill in this questionnaire were on average 57.0 years old and 54.3% women whereas those who completed the SUS were 56.0 years old and 51.5% female. An average SUS of 72.0 shows good acceptance of the app [20] (Table 3). Most users

**Table 1. Participant demographics–distribution of gender and age.**

| | | | Age | | | | |
|---|---|---|---|---|---|---|---|
| | | *n* (%) | Median | Q1 | Q3 | Minimum | Maximum |
| | Total | 313 (100.0) | 52.5 | 43.0 | 62.0 | 24.0 | 76.0 |
| Gender | Male | 148 (47.6) | 54.0 | 46.8 | 62.0 | 24.0 | 76.0 |
| | Female | 163 (52.4) | 52.0 | 40.0 | 60.5 | 26.0 | 76.0 |
| | Missing | 2 | - | - | - | - | - |

Q1: First Quartile, Q3: Third Quartile.

found the system easy to use and felt confident using it. However, less than half of the participants (47.6%) stated they would like to use PIA frequently (S3 Table).

We evaluated several models in the linear regression analysis to determine factors (age group, gender, technology readiness score, app system used) that might influence user acceptance of 97 participants without missing values. In the univariable models, only the technology readiness had a significant influence on user acceptance (S4 Table), so we did not conduct a multivariale analysis. Participants with higher technology readiness scores also had a higher SUS. Still, the model explains only 4.3% of the variation in the data (Table 4).

### Compliance

**Questionnaires.** Users answered 4,237 of the 10,305 weekly health questionnaires (41.1%) that could have been answered between 18 October 2019 and 31 March 2021 in case of perfect compliance of each participant (Fig 4).

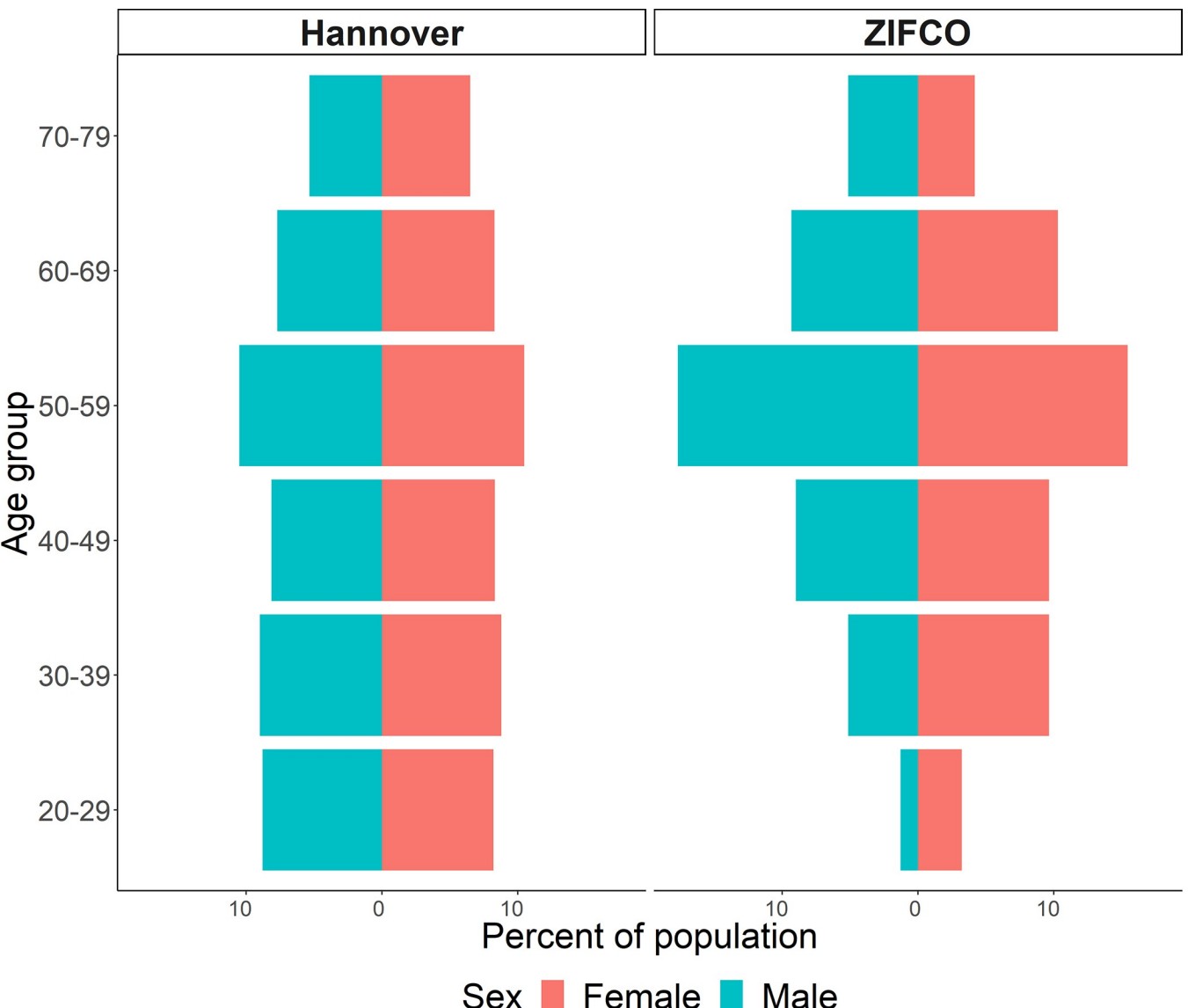

**Fig 3. Comparison of the study population with the source population of Hannover, Germany.** Official population numbers were taken from the county office, reporting date 31.12.2020.

**Table 2. Distribution of app usage in total and across gender and age groups.**

| | | | | App system | | | |
| --- | --- | --- | --- | --- | --- | --- | --- |
| | | | Total | Android | iOS | Web | Multiple |
| | | | *n* (%) | *n* (%) | *n* (%) | *n* (%) | *n* (%) |
| Total | | | 313 (100.0) | 134 (42.8) | 124 (39.6) | 30 (9.6) | 25 (8.0) |
| | | Age group | | | | | |
| Gender | Female | Total | 163 (100.0) | 72 (44.2) | 70 (42.9) | 8 (4.9) | 13 (8.0) |
| | | <40 | 40 (100.0) | 16 (40.0) | 22 (55.0) | 0 (0.0) | 2 (5.0) |
| | | 40–60 | 82 (100.0) | 36 (43.9) | 37 (45.1) | 4 (4.9) | 5 (6.1) |
| | | >60 | 41 (100.0) | 20 (48.8) | 11 (26.8) | 4 (9.7) | 6 (14.6) |
| | Male | Total | 148 (100.0) | 60 (40.5) | 54 (36.5) | 12 (8.1) | 22 (14.9) |
| | | <40 | 20 (100.0) | 9 (45.0) | 8 (40.0) | 1 (5.0) | 2 (10.0) |
| | | 40–60 | 85 (100.0) | 36 (42.3) | 34 (40.0) | 8 (9.4) | 7 (8.2) |
| | | >60 | 43 (100.0) | 15 (34.9) | 12 (27.9) | 13 (30.2) | 3 (7.0) |
| | Missing | | 2 (100.0) | 2 (100.0) | 0 (0.0) | 0 (0.0) | 0 (0.0) |

Fifty percent of questionnaires (2,123/4,237) were submitted on the day of presentation, i.e. Monday (S1 Fig) and 25.4% between 3pm and 5pm (1,075/4,237), which was the most frequent time of submission. Participants that reached good compliance (at least 75% of the weekly health questionnaires answered; reached overall by 44.4% of all participants) were enrolled for a median of 16.0 weeks at the time of analysis. In comparison, those with poor compliance were enrolled for a median of 57.0 weeks. Overall median compliance was 55.7% (Table 5).

In the multifactorial logistic regression with a binomial response approach, we included 247 participants without missing values. In the resulting model the independent variables gender, age, technology readiness and weeks since registration were significant, as well as interactions between being female and over 60 years old. Comparisons of different models can be found in S5 Table. When considering interaction of age and gender, the odds ratio for compliance increases to 2.17 if a person is over sixty and female, while there is no gender difference for the age group 40 to 60. In the age group under 40, the odds to respond are higher for men (Table 6).

**Table 3. Distribution of the system usability score in total and across gender and age groups.**

| | | | | System Usability Score | | |
| --- | --- | --- | --- | --- | --- | --- |
| | | | *n* | Mean (95%-CI) | Minimum | Maximum |
| Total | | | 99 | 72.0 (68.9–75.2) | 27.5 | 100 |
| | | Age group | | | | |
| Gender | Female | Total | 51 | 71.2 (67.0–75.1) | 27.5 | 97.5 |
| | | <40 | 13 | 70.6 (65.2–76.5) | 57.5 | 97.5 |
| | | 40–60 | 21 | 68.5 (63.7–73.2) | 45.0 | 97.5 |
| | | >60 | 17 | 75.0 (64.9–83.8) | 27.5 | 95.0 |
| | Male | Total | 47 | 73.0 (68.1–77.7) | 35 | 100 |
| | | <40 | 7 | 68.2 (54.3–81.4) | 37.5 | 95.0 |
| | | 40–60 | 22 | 74.8 (67.5–81.7) | 35.0 | 97.5 |
| | | >60 | 18 | 72.6 (65.4–79.6) | 37.5 | 100.0 |
| | Missing | | 1 | - | - | - |

CI: Confidence Interval.

**Table 4. Results for the univariable linear regression model for outcome system usability score ($n = 97$).**

| Predictor | | Estimate (95% CI) | $p$-value | Adjusted $R^2$ | AIC |
|---|---|---|---|---|---|
| **Technology readiness score** | | 10.66 (1.45–19.86) | 0.024 | 0.043 | 811.3 |

CI: Confidence Interval, AIC: Akaike Information Criterion.

To examine the adequacy of the model, we explored its corresponding Pearson residuals. While the mean (0.02) is close to 0 as expected, the variance of the Pearson residuals (12.75) is far higher than assumed in the model (it should be close to 1). The visual examination (S2 Fig) showed that this is probably due to time dependent pattern in the data. The variance of the residuals is much lower in the first weeks since registration than later.

**Nasal swabs.** Fifty of the 64 participants (78.1%) that reported symptoms related to respiratory infections sent in their nasal swabs at 112 out of the 145 times (77.2%) they were requested to do so (Table 7).

## Discussion

### Principal findings

We found a good acceptance of PIA (average SUS = 72.0) that was not statistically linked to gender, age, or which app system participants used, but to the technology readiness of the participants. The 44.4% of users that reached a good compliance for submitting weekly health questionnaires (at least 75% of questionnaires submitted) were enrolled for a median time of 16.0 weeks, compared to those with poor compliance who were enrolled for 57.0 weeks, showing that compliance decreased the longer individuals participated. Being female, of younger

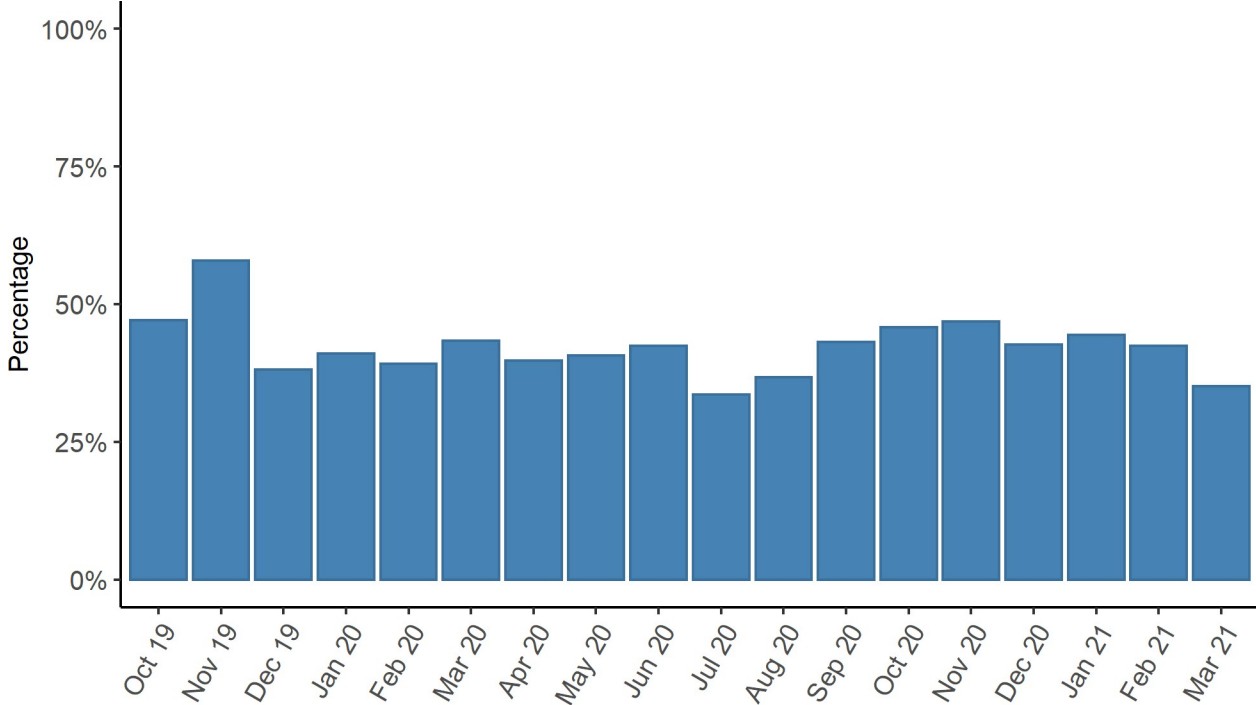

**Fig 4. Percentage of submitted weekly health questionnaires (18 October 2019–31 March 2021, $n = 4{,}237$).** Due to technical issues, PIA was not available from 13 July to 19 July 2020 and from 15 March to 17 March 2021.

**Table 5. Distribution of compliance as proportion of all possible weekly questionnaires to be filled out in total and in dependence on gender, age group and weeks since registration.**

| | | | | Compliance (%) | | | | |
|---|---|---|---|---|---|---|---|---|
| | | | *n* | Median | Q1 | Q3 | Minimum | Maximum |
| **Total** | | | 313 | 55.7 | 8.2 | 100.00 | 0.0 | 100.0 |
| | | **Age group** | | | | | | |
| **Gender** | **Female** | Total | 163 | 43.8 | 9.0 | 98.1 | 0.0 | 100.0 |
| | | <40 | 40 | 29.3 | 5.8 | 62.0 | 0.0 | 100.0 |
| | | 40–60 | 82 | 55.8 | 14.2 | 100.0 | 0.0 | 100.0 |
| | | >60 | 41 | 41.0 | 8.2 | 85.3 | 0.0 | 100.0 |
| | **Male** | Total | 148 | 83.8 | 6.5 | 100.0 | 0.0 | 100.0 |
| | | <40 | 20 | 85.3 | 40.0 | 100.0 | 0.0 | 100.0 |
| | | 40–60 | 85 | 88.9 | 18.6 | 100.0 | 0.0 | 100.0 |
| | | >60 | 43 | 52.9 | 3.6 | 98.3 | 0.0 | 100.0 |
| | **Missing** | | 2 | - | - | - | - | - |
| **Weeks since registration*** | **1–11** | | 107 | 100.0 | 33.3 | 100.0 | 0.0 | 100.0 |
| | **12–23** | | 49 | 78.9 | 26.7 | 100.0 | 0.0 | 100.0 |
| | **24–36** | | 42 | 69.2 | 38.4 | 94.9 | 0.0 | 100.0 |
| | **54–65** | | 51 | 36.8 | 8.4 | 87.4 | 0.0 | 100.0 |
| | **66–77** | | 64 | 11.6 | 1.1 | 49.3 | 0.0 | 100.0 |

Q1: First Quartile, Q3: Third Quartile.

*Closure of study center in response to COVID-19 pandemic resulted in weeks where no participants were enrolled, therefore weeks 37 to 53 are missing.

age and being enrolled for a longer time also significantly lowered the chances to respond. However, women over 60 years had a higher chance to respond in comparison to women younger than 60 years, while men and women aged 40 to 60 had similar chances for good compliance. For participants under 40, the chances for compliance were higher for men than for women.

**Table 6. Results for the logistic regression model for outcome compliance (*n* = 247).**

| Predictor | | Odds ratio (95% CI) | Global *p*-value | Local *p*-value | Log Likelihood Ratio* (*p*-value) | AIC |
|---|---|---|---|---|---|---|
| **Gender** | **Male** | Reference | <0.0001 | | 1059.5 (<0.0001) | 4096.98 |
| | **Female** | 0.39 (0.31–0.49) | | <0.0001 | | |
| **Age group** | **<40** | Reference | <0.0001 | 0.0001 | | |
| | **40–60** | 1.51 (1.22–1.85) | | <0.0001 | | |
| | **>60** | 1.62 (1.31–1.99) | | <0.0001 | | |
| **Technology Readiness Score** | | 1.53 (1.30–1.79) | - | <0.0001 | | |
| **Weeks since registration** | **1–11** | Reference | <0.0001 | <0.0001 | | |
| | **12–23** | 0.33 (0.22–0.47) | | <0.0001 | | |
| | **24–36** | 0.30 (0.21–0.43) | | <0.0001 | | |
| | **54–65** | 0.17 (0.11–0.23) | | <0.0001 | | |
| | **66–77** | 0.09 (0.06–0.13) | | <0.0001 | | |
| **Gender: Age group** | **Male < 40** | Reference | - | | | |
| | **Female 40–60** | 1.02 (0.78–1. 35) | | 0.8568 | | |
| | **Female >60** | 2.17 (1.64–2.87) | | <0.0001 | | |

CI: Confidence Interval, AIC: Akaike Information Criterion.

*Final model tested versus baseline model.

**Table 7. Distribution of compliance as proportion of all possible nasal swabs to be submitted in total and across gender and age group.**

| | | | Participants that were requested to send in a nasal *swab* at least once during the study *(n)* | Participants that sent in at least one nasal swab (n) | Compliance with nasal swabs (%) |
|---|---|---|---|---|---|
| **Total** | | | 64 | **50** | 78.1 |
| | **Age group** | | | | |
| **Gender** | **Female** | Total | 31 | **28** | 90.3 |
| | | <40 | 9 | **8** | 88.9 |
| | | 40–60 | 15 | **13** | 86.7 |
| | | >60 | 7 | **7** | 100.0 |
| | **Male** | Total | 33 | **22** | 66.7 |
| | | <40 | 7 | **7** | 100.0 |
| | | 40–60 | 21 | **12** | 57.1 |
| | | >60 | 5 | **3** | 60.0 |

## Interpretation

Existing studies on factors influencing user acceptance of systems are heterogeneous. Some studies show no substantial differences in perception of user experiences between men and women [21, 22]. Other studies indicate that technology adoption varies by age and gender, e.g. the likelihood of good user acceptance and willingness of use was higher among men or there were gender specific perceptions [23, 24]. Our findings regarding the connection between technological readiness and user acceptance go in line with studies which show that performance expectancy and smartphone experiences are influencing factors in terms of user acceptance [25]. However, low goodness of fit in our model suggests that there are also other factors influencing user acceptance that should be addressed in further research.

Only few studies examine factors that influence compliance. These studies do address social factors such as e.g. education and income rather than age and gender differences [26]. Comparing compliance across eResearch systems is difficult because of the different nature and purposes of the systems, different definitions of compliance, as well as differences in what is considered good or poor compliance. For example, the compliance of a system used for chronic disease management may depend on the severity of the disease. Heterogeneous results imply that user acceptance and compliance depend on multiple factors such as personality dimensions e.g. interests of the individual, subjective norms and attitudes as well as perceived usefulness and system specific dimensions like ease of use [27, 28].

Considering the generally low participation rates that have been reported in epidemiological studies for the last years [29, 30], suboptimal compliance may be dissatisfactory for longitudinal studies but is indicative of a general trend. Furthermore, earlier studies that reported high compliance with web-based questionnaires [3, 31] might have had the advantage of having a fixed study period that may motivate participants to complete the study, which ZIFCO does not have. Participants in ZIFCO are asked to participate for as long as possible, ideally for years. This is also evident in our findings that participants with shorter enrollment had better compliance than those who were enrolled for a longer period. Contrary to this assumption, the German internet-based participatory surveillance system GrippeWeb reported high participation in a sample over the span of one year despite not having a fixed study period. However, they also concluded that participation rates were positively influenced by the inclusion of an incentive system [32]. Using monetary incentives to increase compliance, though effective [33, 34], are not feasible in large long-term cohort studies. Integrating principles of gamification e.g. through implementation of feedback mechanisms provide an alternative to this and have

been used successfully in other eHealth applications [35, 36]. Adapting them is the focus of the future development in PIA and might positively influence compliance in the future particularly in younger study participants. In addition, the usability of PIA should be continuously improved to simplify handling of the app. Better compliance with nasal swab self-sampling than with submission of weekly health questionnaires provide a first indication that sending participants' feedback on their laboratory results might be feasible to motivate participants to use PIA as shown elsewhere [37], but more research is needed when more participants are enrolled over time.

## Limitations

Our study so far only includes 313 active users of PIA. Although men and women participated in similar numbers, older participants are overrepresented, reflecting the study population of the NAKO [17]. Study nurses sometimes reported insufficient time to additionally recruit for ZIFCO during the standard protocol of the NAKO as well as occasional technical problems or material shortage influencing recruitment. Consequently, fewer ZIFCO participants were included than would have been possible in terms of willingness to participate. Regarding the user acceptance, it is important to consider that NAKO participants who agree to use a mobile or web application for epidemiological surveys (participants who agreed to take part in ZIFCO), might be more technologically affine than the general study population.

Furthermore, taking more factors into account that might influence compliance, like status of employment [31], could explain more of the variance in the model. The current circumstances due to COVID-19 might additionally have affected user behavior and thus not represent user compliance during non-pandemic times.

Residual analysis revealed that a different modelling approach for the compliance should be considered as the compliance in the first weeks after registration is nearly perfect and then disperses. One hypothesis is that there are different types of responders as postulated by Akmatov et al. (2014) which causes heterogeneity. Still, our model helps to understand relationships between compliance and the covariates at hand. A more in-depth analysis of how the duration of participation is correlated with the compliance for the influence of e.g. gender and age on compliance should be done in following studies.

## Conclusions and recommendations

Our study showed that PIA is an acceptable eResearch system and suitable within an epidemiological research context but that methods to increase compliance are strongly indicated for successful long-term adherence. Current development of PIA that places emphasis on user engagement and gamification might help incentivize regular and long-term participation in the future and with more participants enrolled over time we will be able to review und refine our findings.

## Supporting information

**S1 Table. Distribution of technology readiness score (range 1 to 5) in total and across gender and age group.** Q1: First Quartile, Q3: Third Quartile.
(DOCX)

**S2 Table. Answers to technology readiness questionnaire (n = 258).**
(DOCX)

**S3 Table. Answers to user acceptance questionnaire (SUS) (n = 104).**
(DOCX)

**S4 Table. Results for the univariate linear regression models for outcome system usability score.** CI: Confidence Interval, AIC: Akaike Information Criterion.
(DOCX)

**S5 Table. Results for the multivariate logistic regression models for outcome compliance.** AIC: Akaike Information Criterion. *Tested against baseline model.
(DOCX)

**S1 Fig.**
(TIFF)

**S2 Fig.**
(TIFF)

## Acknowledgments

This project was conducted with data from the German National Cohort (NAKO) (www. nako.de). We thank all participants who took part in the NAKO study and the staff in this research program.

## Author Contributions

**Conceptualization:** Jana-Kristin Heise, Stefanie Castell.

**Data curation:** Julia Ortmann, Jana-Kristin Heise.

**Formal analysis:** Julia Ortmann.

**Funding acquisition:** Stefanie Castell.

**Investigation:** Felix Jenniches, Yvonne Kemmling.

**Methodology:** Jana-Kristin Heise, Irina Janzen, Stefanie Castell.

**Project administration:** Jana-Kristin Heise, Stefanie Castell.

**Resources:** Jana-Kristin Heise, Stefanie Castell.

**Software:** Julia Ortmann, Jana-Kristin Heise.

**Supervision:** Cornelia Frömke, Stefanie Castell.

**Validation:** Irina Janzen, Cornelia Frömke.

**Visualization:** Julia Ortmann, Irina Janzen.

**Writing – original draft:** Julia Ortmann.

**Writing – review & editing:** Julia Ortmann, Jana-Kristin Heise, Irina Janzen, Felix Jenniches, Yvonne Kemmling, Cornelia Frömke, Stefanie Castell.

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
