## [Decision Letter · Decision Letter 0]

24 Oct 2022

PONE-D-22-02047Suitability and User Acceptance of the eResearch System “Prospective Monitoring and Management App (PIA)” - the example of an epidemiological study on infectious diseasesPLOS ONE

Dear Dr. Castell,

Thank you for submitting your manuscript to PLOS ONE. After careful consideration, we feel that it has merit but does not fully meet PLOS ONE’s publication criteria as it currently stands. Therefore, we invite you to submit a revised version of the manuscript that addresses the points raised during the review process.

We recognize that this decision was delayed due to reviewer availability.  As Academic Editor, I undertook the send review, as this falls in my area of expertise. While one reviewer recommended a major revision, I did not think the comments required major edits.  There are minor revisions that we recommend for a follow up submission. 

We look forward to receiving your revised manuscript.

Kind regards,

Kimberly Page, PhD, MPH

Academic Editor

PLOS ONE

Journal Requirements:

Additional Editor Comments (if provided):

Reviewers' comments:

Reviewer's Responses to Questions

**Comments to the Author**

1. Is the manuscript technically sound, and do the data support the conclusions?

Reviewer #1: Partly

Reviewer #2: Yes

2. Has the statistical analysis been performed appropriately and rigorously? 

Reviewer #1: Yes

Reviewer #2: Yes

3. Have the authors made all data underlying the findings in their manuscript fully available?

Reviewer #1: Yes

Reviewer #2: No

4. Is the manuscript presented in an intelligible fashion and written in standard English?

Reviewer #1: Yes

Reviewer #2: Yes

5. Review Comments to the Author

Reviewer #1: The authors present a clear, easy to follow analysis on the study and the topic of the study is interesting, especially given the potential of using digital tools for ongoing surveillance is a much talked about but relatively-less researched area.

In the conclusions, there are 2 major areas where I would like to see more analysis--

1. What inference is possible to draw here between the representativeness of the study participants who responded to user acceptance rates and the population of intended use? Since the conclusions are talking about a general use of the tool for surveillance data collection, it would be important to understand this.

2. What was the reason for non-completion of questionnaires/missing information and does this have an implication on the overall conclusions?

Depending on the responses to the above, the next steps recommended in the study should be modified.

Reviewer #2: This paper describes acceptability and use compliance in association with an 'app' (Prospective Monitoring and Management App [PIA]) deployed to assess respiratory illness. The app was tested among participants of a German Cohort. System Usability Scale was used to assess user acceptance four months after introduction of PIA. Compliance was assessed as the proportion of weekly surveys collected over time. As well, compliance with nasal self-swabbing was assessed among participants these samples were collected from. Results show good acceptance and compliance overall. Variations by sex and age were observed. Technology readiness was a strong indicator of acceptability. The goals of the paper are well described and the methods and analyses used are appropriate. Results and data interpretation are cohesive. Overall this paper describes a potentially usable method of collecting data remotely, using app or web interface. I think the conclusions and recommendations are also good and it will be interesting to see if this can be deployed more widely.

I suggest a few minor edits and explanations:

1. Line 86: please spell what ARI refers to.

2. Line 121: was there a rationale for the threshold of 75% indicating 'good' compliance?

3. Line 133 and elsewhere (eg., line 180): I am assuming that what the authors describe as 'univariate' model is actually bivariate. Please consider: Univariate statistics summarize only one variable at a time. Bivariate statistics compare two variables.

4. Please describe where the N=313 comes from. It is not in the text (lines 143-148, nor in the flow chart of enrollment and data analysis. The numbers don't quite make sense.

6. PLOS authors have the option to publish the peer review history of their article (what does this mean?). If published, this will include your full peer review and any attached files.

Reviewer #1: No

Reviewer #2: No

---

## [Author Response · Author response to Decision Letter 0]

12 Dec 2022

We sincerely thank the reviewers for their constructive feedback. We revised our manuscript accordingly and think that it improved in clarity. Please see the file "Response for reviewers" where we have answered the comments and questions of the reviewers in detail.

---

## [Editor Report · Decision Letter 1]

19 Dec 2022

Suitability and User Acceptance of the eResearch System “Prospective Monitoring and Management App (PIA)” - the example of an epidemiological study on infectious diseases

PONE-D-22-02047R1

Dear Dr. Castell,

We’re pleased to inform you that your manuscript has been judged scientifically suitable for publication and will be formally accepted for publication once it meets all outstanding technical requirements.

Kind regards,

Kimberly Page, PhD, MPH

Academic Editor

PLOS ONE
---

## [Editor Report · Acceptance letter]

22 Dec 2022

PONE-D-22-02047R1 

Suitability and User Acceptance of the eResearch System “Prospective Monitoring and Management App (PIA)” - the example of an epidemiological study on infectious diseases 

Dear Dr. Castell:

I'm pleased to inform you that your manuscript has been deemed suitable for publication in PLOS ONE. Congratulations! Your manuscript is now with our production department. 

Kind regards, 

on behalf of

Dr. Kimberly Page 

Academic Editor

PLOS ONE